# Co-Creating and Evaluating an App-Based Well-Being Intervention: The HOW (Healthier Outcomes at Work) Social Work Project

**DOI:** 10.3390/ijerph17238730

**Published:** 2020-11-24

**Authors:** Jermaine M Ravalier, Elaine Wainwright, Nina Smyth, Oliver Clabburn, Paulina Wegrzynek, Mark Loon

**Affiliations:** 1Newton Park Campus, School of Science, Bath Spa University, Bath BA29BN, UK; e.wainwright@bathspa.ac.uk (E.W.); p.wegrzynek2@bathspa.ac.uk (P.W.); m.loon@bathspa.ac.uk (M.L.); 2School of Social Sciences, Psychology, University of Westminster, London W1W 6UW, UK; n.smyth@westminster.ac.uk; 3Wellcome Centre for Cultures and Environments of Health, Streatham Campus, University of Exeter, Exeter EX4 4QH, UK; o.clabburn@exeter.ac.uk

**Keywords:** well-being, working conditions, intervention, app, stress, mental health

## Abstract

Stress and mental health at work are the leading causes of long-term sickness absence in the UK, with chronically poor working conditions impacting employee physiological and psychological health. Social workers play a significant part in the fabric of UK society, but have one of the most stressful occupations in the country. The aim of this project was to work with UK social workers to co-develop, implement, and evaluate a series of smartphone-based mental health initiatives. A Participatory Action Research (PAR) approach, consisting of semi-structured interviews and focus group and steering group discussions, was utilized to design the mental health and well-being interventions. Study efficacy was evaluated via a pre- and post-intervention survey and post-intervention semi-structured interviews. Interventions developed were psycho-educational, improved top-down and bottom-up communication, and provided access to a Vocational Rehabilitation Assistant for those struggling and at risk of sickness absence. Six months following dissemination, surveys demonstrated significant improvements in communication, and mean score improvements in four other working conditions. This project, therefore, demonstrates that co-developed initiatives can be positively impactful, despite post-intervention data collection being impacted by COVID-19. Future studies should build upon these findings and broaden the PAR approach nationally while taking a robust approach to evaluation.

## 1. Introduction

Employee well-being should be an important consideration for employees, employers, and policy makers alike [1]. Indeed, work stress, depression, and anxiety are among the biggest reasons for both long- and short-term sickness absence in the United Kingdom (UK) [2]. Work stress, therefore, costs individuals in terms of psychological and physiological health and well-being [3]. It costs employers due to having high levels of burnout [4], turnover of intentions [1], and job satisfaction [5], among other things. Employee well-being should also be a consideration for policy makers because research has demonstrated that patients in clinical settings have higher levels of morbidity and mortality, which have worsened employee well-being [6]. In the UK, Health and Social Care (H&SC) workers have among the highest levels of stress and mental health-related sickness of all occupations [3].

H&SC workers are key responders across the world to the COVID-19 pandemic [7]. Indeed, the World Health Organization [8] describes H&SC worker mental health as being as important as physical health during the COVID-19 crisis, impacting patient mortality and morbidity outcomes [6,9] among other things. However, the Department for Health and Social Care (DHSC) acknowledges that little mental health support is available, and evidence has demonstrated that COVID-19 has added significantly to the difficult working conditions already faced by these key workers [10]. The aim of this project was to co-design and evaluate the efficacy of a series of mental health and well-being initiatives for UK social workers, with initiative dissemination and evaluation occurring during the COVID-19 pandemic and UK national lockdown.

### 1.1. Stress and Health

In the UK, stress, depression, and anxiety are responsible for over 15 million working days lost per year, equivalent to 57% of all days lost due to sickness absence [3]. As such, these three conditions are the biggest cause of long-term sickness absence (i.e., lasting four weeks or more) and second to short-term absences such as cold and flu [2]. It is clear, therefore, that chronic stress at work is harmful to both the physiological and psychological well-being of staff. Similarly, a systematic review by Hassard and colleagues [11] found that work-related stress costs from between $221 million and $580 million per year in Australia, to $4.36 billion per year in France, and $187 billion in the USA.

A high level of work stress (and, in particular, low levels of control and peer support and high workload) has been shown to play an important role in cardiovascular mortality [12] and cardiovascular diseases [13]. It has also been linked to metabolic syndrome (a risk factor for the development of type 2 diabetes) [14], depressive symptomology [15], and other negative mental and physical health outcomes [16].

Chronic work stress, therefore, affects employing organizations in terms of burnout [17], staff turnover [18], absenteeism [5], presenteeism [1], and the costs associated with employee sickness absence [16]. Importantly, the effects of chronic stress can also be seen on the patients/service users whom frontline H&SC workers support. For example, West and Dawson [6] identified a relationship between employee engagement and patient mortality and morbidity. Similarly, poor well-being in social workers has been related to increased numbers of errors [19], suicide risk [20], and worsened practice [1] and, thus, has the potential to cause negative consequences for service users.

### 1.2. Working Conditions and Stress

The job demands-resources (JDR) model [21] is one of the most widely adopted theories of work stress. It was initially developed for application in the area of burnout, which is a specific construct characterized by features such as emotional exhaustion and feeling personally unaccomplished [21]. It has since been widely applied to various workplace well-being considerations, including the development of health improvement interventions [22].

The JDR suggests that work conditions can be characterized as either resources (which detract from the experience of stress at work) or demands (which contribute to the stress experience). Workplace resources include physical, psychosocial, and/or organizational areas of the job that can support the undertaking of work [23]. Having sufficient resources, therefore, supports the positive well-being of employees. Alternatively, demands are any aspect of the job that require cognitive or emotional effort, and thus, have the potential to add to the experience of stress. Should there be an imbalance between these demands and resources experienced through work, then stress-related strain and associated outcomes may occur [24,25].

In 2004, the UK Health and Safety Executive (HSE) [26] released a set of management standards (also known as psychosocial working conditions) that could be used by researchers and employers to investigate and promote better psychosocial working conditions [26]. These management standards have been demonstrated to adequately explain and support the JDR [1]. The standards were developed based on a review of the existing literature and suggest that employee stress can occur if working conditions remain in a poor state for a chronic period of time [26]. The Management Standards’ working conditions’ model suggests that the psychosocial hazards that contribute toward the experience of stress can be categorized into seven distinct factors: demands, control, peer support, relationships, managerial support, role, and change [26]. The HSE also released a set of diagnostic tools (the Management Standards Indicator Tool, MSIT) so that organizations could be compared against national standards [26], with Edwards and Webster [27] providing UK-wide benchmark scoring for a short-form, 25-item version of the MSIT (HSE, UK).

The management standards approach, and MSIT, have been demonstrated to be predictive of psychological well-being and related work outcomes across a number of employment sectors such as social workers [1,5], the police [28], and teachers [29]. Indeed, using the management standards approach, and MSIT, Ravalier [5] demonstrated that UK social workers are exposed to worse working conditions than most other UK occupations, and work on average 11 h per week more than they are contracted to do.

### 1.3. Child and Family Social Workers

In the UK, social work is understood to be one of the most stressful occupations [5]. Indeed, the Health and Social Care sectors combined have the highest levels of stress sickness absence of all occupations [3]. Research has also demonstrated that social workers have among the worst working conditions of all occupations in the country, with working conditions as measured in the Management Standards worse than 90%–95% of the UK average, with these psychosocial working conditions subsequently influencing well-being, turnover intentions, and job satisfaction [5].

Well-being is a particularly important focus for Child and Family social workers (C&FSWs). The role of the C&FSW is to work alongside other professionals in order to ensure that children receive the appropriate health, education, and care services [30]. C&FSWs, therefore, play a vital role in the lives of the families with whom they work. However, it has been shown that Child and Family social workers work on average 11 h more per week than they are contracted to—more than any other social work role [5].

### 1.4. Aim and Research Questions

Considering the difficulties highlighted concerning the mental health and well-being of UK social workers and the impact that chronically stressful working conditions can have on them, the aim of this project was to co-design, disseminate, and evaluate a series of stress management interventions with and for UK social workers.

**Research Question 1:** to co-design and disseminate a series of app- and toolkit-based mental health and well-being initiatives for UK social workers.

**Research Question 2:** to evaluate the efficacy of the initiatives via a pre- and post-intervention survey of working conditions and well-being.

**Research Question 3:** to evaluate efficacy of the interventions with post-intervention semi-structured interviews.

## 2. Methods

### 2.1. Design

This paper reports the findings of a mixed-methods approach to the co-design, dissemination, and evaluation of a series of well-being initiatives for C&FSWs employed in seven local authorities (LA) based in the Midlands, South West, and North East of the UK. In the UK, the majority of social workers are employed within LAs [31]. LAs are public-sector organizations that are predominantly funded by central government via public taxation. The principal social worker (PSW) in each organization was approached by JR because of their geographic spread and because they represent either a rural (four LAs) or city (three LAs) locality. Ethical approval was gained from the Bath Spa University research ethics board. The ethical clearance code is: ESRCJR.

Figure 1 below outlines the stages of data collection and intervention rollout across the project. There were five separate but interrelated sets of data collection. A baseline survey (January–February 2019) was conducted to set a baseline and allow post-intervention evaluation. Qualitative interviews (March 2019) and focus groups (April and May 2019) were undertaken in order to inform intervention development. Following development, six months were allowed for the rollout and bedding in of interventions [32]. Post-intervention surveys (February and March 2020) and interviews (March and April 2020) evaluated the efficacy of interventions and impact on health and well-being. All C&FSWs employed in each LA were invited to participate in each stage of the project via email. This was sent on behalf of the research team by the PSW. Participant informed consent was gathered within each distinct stage of the study.

A Participatory Action Research (PAR) approach was taken in order to work with social workers in developing the well-being initiatives. PAR seeks to ensure that project beneficiaries play at least some part in the whole research process, meaning knowledge is co-created between researchers and participants [33]. Participants’ knowledge and expertise in their own social situation was, therefore, used to develop well-being interventions to be disseminated through a smartphone app and associated well-being toolkit. Researchers, therefore, did not enter the research phase assuming that they knew which interventions should be put in place within organizations. Rather, they relied on the expertise and experience of those who were working in the intended job role [33].

Interventions were developed through a series of interviews and related focus groups with C&FSW staff. Interviews allowed participants to outline interventions that could be provided through a smartphone app, which they would like to see available within their organization. Focus groups then allowed further exploration of these interventions, and steering group discussions put these into a workable and feasible structure.

### 2.2. Methods and Materials

#### 2.2.1. Qualitative Materials

The PAR process comprised of a number of separate but interrelated steps. A thematic, analytic framework was used to guide data collection and analytic processes [34,35]. Firstly, a series of individual semi-structured interviews were undertaken in which all social worker professionals were invited via email. Interested parties were given information on this part of the study and formally consented. Each interview began with a reminder of ethical considerations and consent checking. The aim of the interviews was threefold: to identify the key everyday stressors in the role, to discuss and imagine the type of stress management intervention that could be put into place to alleviate some of these stressors, and to discuss the ways in which these could be presented through an app. Interview questions subsequently focused upon the difficulties associated with the social worker role (such as, ‘Has your job become more or less manageable over the last 12 months or so?’), examples of best practice mental health and well-being support they have experienced at work (for example, ‘Can you give me some examples of the best support you have received for mental health in this or any other organization?’), and what support they would like to see put into place both in their organization and through the smartphone app and toolkit (‘If you were rolling out new well-being support in your organization, how would you do it?’).

Next, the aim of focus group discussions was to determine which of the stress management interventions to emerge from the interviews were feasible and workable within the project and interventions. Focus groups were run virtually, via the GoToMeeting online platform, with participants in each focus group being a mix of employment grades (all were social workers or social work team leaders), with each of the seven LAs represented in every focus group. Interviews and focus groups took place with an opportunity sample of individuals who were able to participate in both forms of data collection if they wished. Every focus group began by stating the group ground rules, followed by a series of questions based on the best practice findings from the semi-structured interviews, as well as space allowed for individuals to put forth and discuss newly emergent topics [36].

Finally, steering group discussions were held every three to four months with senior social workers and colleagues from across the seven LAs who had an interest in colleague well-being. The aim of these discussions was to discuss the progress of the project, ensure collaborative relationships were ongoing, and to evaluate the feasibility of implementing initiatives within their organization. These discussions took place between at least one member of the research team and two organizational representatives from each LA (at least one senior social worker) at key points in the project to ensure successful design and delivery of interventions.

#### 2.2.2. Instruments and Scales

A two-pronged approach to evaluation was undertaken: firstly, a pre- and post-intervention survey (collected via the Qualtrics.com online platform) and, secondly, a series of post-intervention semi-structured interviews. The survey consisted of the same measures, of working conditions and well-being, of demographics as pre-intervention. However, questions on frequency (and feedback) of app and toolkit usage were only asked in the follow-up survey.

Working conditions were measured using the MSIT, a 25-item tool that measures seven areas of the workplace (demands, control, management support, peer support, relationships, role, and change), with psychometric validity and reliability previously demonstrated [27]. Responses were given on a four-point scale, ranging from 1–5. Items for the demands’ and relationships’ dimensions were reverse-scored. Items were summed and higher scores equated to better working conditions.

Mental health and well-being were measured using the General Health Question (GHQ), 12-item version (GL Assessment, UK). The GHQ is a measure of general psychological health and can be used as a screening tool for psychological morbidity [37]. Responses were given on a four-point scale, ranging from 0 to 3. Scores were calculated using the simple Likert scoring method by summing the items. Possible scores could range between 0–36, with higher scores on the GHQ indicating worse well-being. Again, the GHQ has been shown to be valid and reliable for use in occupational settings [38,39,40]. The MSIT and GHQ have been shown to be valid for use together in previous studies [41]. Cronbach’s alpha was good (ranging between 0.7–0.9) for all scales, except the relationships’ dimension of the MSIT.

The evaluative interviews aimed to provide feedback specifically about the app intervention. As such, interviews reported here were conducted with confirmed app users, and questions focused on what worked well with the app, what needed to be improved, and how such improvements could be achieved. The interview schedule was, therefore, exploratory in nature to investigate this topic.

### 2.3. Analytical Approach

Qualitative interviews were analyzed using a data-driven Thematic Analysis (TA) approach. As a method of analysis, TA seeks to draw general themes from across semi-structured interviews (among other qualitative data techniques). It, therefore, allows the identification, analysis, description, and reporting of themes found across a series of interviews [42]. The six-step approach, advocated by Braun and Clarke [34,35], was taken to conduct analysis. For all qualitative data from both interviews and focus groups, audio recordings were transcribed verbatim then anonymized. Research team members immersed themselves in the data through repeated transcript readings and note taking, creating initial codes that organized the data into salient segments. Codes were combined to create broader themes. A pragmatic approach to theme development was taken. Therefore, findings from the pre-intervention interviews were focused on intervention development, and findings from the post-intervention evaluative interviews focused upon feedback on app usage. Iterative phases of data collection and analyses were undertaken until the researchers judged insights had been sufficiently captured, enabling exemplification of codes and themes.

Focus groups were analyzed and member checked during each session [36]. One member of the research team facilitated each discussion, with that member also noting all ideas and topics to emerge from the work on a virtual white board. The final 30 min of each session were subsequently kept solely for discussion of these topics that emerged through the group. This allowed member checking of ideas and approaches, and ensured that the initiatives to emerge from within the project were truly user-led.

Steering group discussions were neither recorded nor analyzed because the aim of the groups was to receive feedback from across each organization as to the latest steps and findings from the organizations. The steering groups, therefore, were important in assessing the feasibility of interventions, but were not a data collection or analysis approach in their own right.

Quantitative data was analyzed using IBM SPSS 22.0 (IBM, Armonk, NY, USA). A one-way, between-subjects’ multivariate analysis of variance was carried out to assess the impact of the intervention on working conditions. The between-subjects’ factor was comprised of three groups: working conditions in respondents pre-intervention and working conditions post-intervention in respondents who were aware of the app and those who were not aware of the app. The dependent variables comprised scores on seven measures of working conditions, all of which were dimensions of the MSIT. Assumptions of homogeneity of variance-covariance matrices and equality of variance were confirmed (with the exception of the relationships’ dimensions), and moderate correlations were observed among the seven working conditions. Bonferroni correction was applied to take account of type 1 errors. Due to unequal group sizes, with significantly more respondents completing the pre-evaluation forms, we checked that results were consistent on smaller samples randomly selected for respondents completing the pre-evaluation forms.

## 3. Results

### 3.1. Intervention Development: Semi-Structured Interviews and Focus Groups

A series of 19 individual semi-structured interviews were undertaken. This set of interviews lasted, on average, 40 min. The interviews were followed by four focus groups consisting of eight people and lasting 90 min each.

### 3.2. App/Intervention Contents

The app was developed by social workers, for social workers, in each of the seven LAs. Therefore, the contents of the app were developed to be general enough to be applicable to all organizations, while certain sections were also tailored to the needs and offerings of individual organizations. Individuals could only sign into the app by selecting one organization (i.e., their employer) and inputting an organizationally specific password. This also meant that the app could be tailored to individual organizations through a web-based content management system. This allowed either the research team or an organizationally appointed representative to update the app for their own organization. This ensured app content could be regularly updated as needed. This was the same mechanism that allowed push notifications to be sent to all users. The app was developed by an external company (Finesse Management) and took approximately two months to develop.

After first password sign-in (this only needed to be done once), users were automatically taken to their organization’s home page. This home page allowed users to select which part of the app/intervention they wanted to be directed to. Through interviews, respondents suggested that in social work there can exist a negative, blame culture (“i.e., [a non-social work professional apportioning] blame on me, which I was really unhappy with and took it above her”, Interviewee 25). To attempt to redress this often negative culture, the top of the home page also contained a positive message about the work of social workers, taken from a dedicated Twitter account, in which social workers shared positive stories about their work.

The first major section of the app was psychoeducational in nature. Based on the UK HSE [43] suggestions, it outlined ‘*Signs and Symptoms of Stress*’ in themselves, colleagues, and wider teams. Linked to this was the suggestion that, should individuals experience any of these symptoms in themselves, or observe in others, they should refer to organizational sources of support such as occupational health or employee assistance programs. While some did recognize that they understood the signs and symptoms of stress in themselves (“Yeah I think I’m quite good at that. Like from the past I’ve seen people that got quieter. I only get quiet if I’m feeling stressed”. Interviewee 3), others felt that they would be completely unaware if a colleague (or even themselves) were struggling (“To be honest I’d have no idea—I mean people always say that they’re ‘stressed’ or ‘a bit down’. When does this turn into a proper issue?” Interviewee 5). Therefore, this psychoeducational element of the app would help users to understand when either themselves or others were struggling and, thus, could be signposted toward organizational sources of support.

Through both interviews and focus groups, it was suggested that individuals knew that well-being-related events were available (e.g., conferences, workshops, training, exercise classes), but either did not know about them until too close to the event or did not know where the events were taking place (“Well, they could always get people to go into departments and saying right, advertising it and putting it out on bulletins or having it all in one place so we can see what’s happening and when”, Interviewee 3). The app, therefore, had an ‘*Events*’ page that was regularly updated for each organization, as well as push notifications sent to users 72 h prior to any event starting. This meant that all events related to well-being could be advertised in one place, and app users could find information about each event through the app.

A lack of bottom-up communication was also described across the qualitative data collection. Respondents described consultations often seeming as a tick-box exercise, and that when they did respond to management request for input, they were often not responded to (“Not having that space to be heard”, Interviewee 13). To improve upon this, individuals could send (anonymous, if preferred) messages through the app directly to management in their organization regarding how their role or organization could be improved. Management then responded to at least three suggestions each month in their all-staff newsletters, outlining why they had (or had not) acted upon any received feedback and suggestions.

Finally, app users who were struggling with their own mental health and/or well-being due to work, could access a dedicated Vocational Rehabilitation Assistant (VRA). Respondents acknowledged that organizational occupational therapy was an important resource, but it often took time to be able to access it due to a lack of funding: “I think OT [Occupational Therapy] is so necessary to get support in a timely and safe way but not enough resources are given to occupational therapy” (Interviewee 24). The VRA is a trained occupational therapist and, when contacted, sought to work with individual employees and their managers to support the employee to remain in the workplace. Due to GDPR (General Data Protection Regulation) and ethical considerations, the number of self-referrals to the VRA is unknown (and were not asked about in the evaluative portions of the project), but interventions would have included changes to working practices, changes to the working environment, and reducing the number of days worked to support psychological recovery.

### 3.3. Intervention Implementation

Following focus groups, the developed interventions were refined in accordance with the feedback provided and latterly brought to the steering group for additional feedback. As a result, additional changes were made to the interventions, which made them more suitable for each individual participating organization, such as the development of an app ‘storyboard’. This storyboard outlined what pages of the app should look like and, importantly, the interactivity/functionality of the app. This storyboard formed the design basis for the app developer in creating the HOW (Healthier Outcomes at Work) Social Work app. The app was available for use on Android and Apple devices.

The app was disseminated across the seven LAs via a number of all-staff emails and newsletters and promotional events, with these differing strategies aimed at explaining the use/utility of the app, giving a series of light-touch training sessions and, ultimately, encouraging the download and use of the app. First of all, the research team designed a number of promotional posters and fliers. These were delivered to each organization and requested that they were placed in communal areas. Furthermore, a number of business card-sized materials was printed. These were handed out during team meetings and as part of induction packs for new staff. A QR code was present on each type of promotional material that would allow participants to download the app onto their personal smartphones. At least three lunch events were also held in each LA during the first two months of intervention launch. This helped to raise awareness of the app and provide informal training sessions on use. Principal social workers were also given instructions as to how to train others within the organization to use the app. By April 2020, there were 483 individual downloads of the app (meaning 29% of available social workers downloaded it). However, due to GDPR considerations, no other data was collected regarding the usage of the app.

### 3.4. Interventional Evaluation

#### 3.4.1. Pre- and Post-Surveys

Results are presented on evaluations completed pre and post across five local authorities; usable survey responses obtained pre-intervention were 503 and post-intervention were 154. Of those completing the post evaluation survey 56 (36%), respondents reported being aware of the HOW app. Table 1 provides mean and benchmark scoring for the MSIT and GHQ.

A one-way, between-subjects’ ANOVA was conducted to examine differences in the GHQ between respondents completing the pre-evaluation form and respondents who were aware or not aware of the HOW app following implementation of the intervention. Overall, there were significant differences in well-being between the three groups (F (1, 2) = 4.355, *p* = 0.013). Bonferroni post hoc tests revealed significant differences between groups; well-being was significantly worse for respondents who were not aware of the HOW app in comparison with respondents completing the pre-evaluation forms (*p* = 0.026) and those completing the post-evaluation forms post-intervention (*p* = 0.027). Results were consistent when the sample for the pre-intervention respondents was limited to 200 respondents.

A one-way, between-subjects’ MANOVA was conducted to examine changes in working conditions between respondents completing the pre-evaluation form and post-intervention in respondents who were aware or not aware of the HOW app. There was a significant difference between the three groups on the combined dependent variable ‘working conditions’ (F (14, 1266) = 2.116, *p* = 0.009; Wilk’s Lambda = 0.955). Analysis of each individual-independent variable, using a Bonferroni adjusted alpha level of 0.007, showed that there was a significant difference in the working condition ‘change’ between the three groups (F (2, 639) = 9.040, *p* < 0.001). Post hoc tests revealed those who were aware of the app post-intervention reported significantly more control over changes in their organization compared with respondents pre-intervention (*p* < 0.001) and those who were not aware of the app post-intervention (*p* = 0.032). There were no significant differences between groups for the other working conditions. Results were consistent, though there was only a trend for the overall model, when the sample for the pre-intervention respondents was limited to a smaller sample.

Percentile scoring compares the scoring of the working conditions against UK national averages. Several working conditions (demands, control, relationships, role, and change) were worse for Social Workers compared with the national average. Following the intervention, working conditions were improved for these conditions in respondents who were aware of the HOW app.

#### 3.4.2. Evaluative Interviews: Working Well

The aim of the semi-structured evaluative interviews was to understand individual social worker feedback upon usage of the app. Fifteen evaluative interviews, lasting on average 29 min each, were undertaken, although six interviewees had not heard of and/or used the app, and these results reflect upon the nine who did. Iterative rounds of data collection and analysis were conducted, with interviewing continuing through to saturation point within which no new themes were created [44].

For those who did use the app, it was found to be a one-stop shop for everything well-being-related within their organization, thus, a huge advantage. Prior to the project (and, therefore, use of the app), there was no one place in which they could go to find out what could be done to support their well-being and, importantly, what was currently being done to maintain positive health and well-being. Until this point, most well-being-related information was held on organizational intranets. However, these intranets were only accessible while at work on premises, which meant accessing support outside of work could be difficult (although, it was noted that the employee assistance programs were always available for use, regardless of place of access): “it’s always this thing where there are events and stuff happening but we either can’t get to them or by the time we hear about them it’s too late, so it’s good to have it all in one place” (Interviewee 5).

Similarly, the communication function within the app was particularly useful for individuals. They liked receiving information through the app via push notifications and being able to log into the app to see the latest information and news: “it’s good to see when and where things are happening” (Interviewee 10). Furthermore, individuals were able to either anonymously (or not, if preferred) provide feedback to management on anything regarding their job, role, or overall organization. Senior management within organizations would then respond to two or three pieces of this feedback submitted via the app in organizational-wide newsletters each month. This meant that not only was the app useful for providing information, but it also allowed individuals to have their say on how things could be improved for them, their team, and, ultimately, their service users: “so often we have consultations and discussions and then nothing never happens. At least this way if we did send stuff up we know we’ve been listened to, even if it’s not used.” (Interviewee 9).

#### 3.4.3. Needing Improvement

There were some areas for improvement that came through the semi-structured interviews. First of all, the dissemination strategy could be improved. For example, many employees had smartphones and tablets that were provided by their employing organization. Due to IT security measures, often individuals could not download the app themselves onto their work phones: “we just don’t have admin rights, so can’t download things on our work phones” (Interviewee 4). We should, therefore, have worked more closely with organizational IT departments to ensure that the app was rolled out and installed onto all staff smartphones. Additionally, there should have been more dissemination of the project, app, and toolkit across team meetings, for example: “these are the kind of things that need to be raised in team meetings, more kind of, ‘This isn’t a choice, this needs to happen’” (Interviewee 2).

A second issue related to the VRA. Many liked the idea of having access to a dedicated, virtual, and independent VRA. However, because it was a completely new innovation within each of the organizations, it was met with a level of skepticism as it had not been seen or used before. For example, interviewees wondered whether the VRA was truly independent of their organization, their occupational health system, and/or their employee assistance program: “The Vocational Rehab person was a good idea, but talking to colleagues we weren’t sure how independent they were, so didn’t want to use them. Like, would they go back to our bosses like Occy [occupational] Health do?” (Interviewee 11). Again, with greater awareness of the app and its contents, much of this skepticism would be overcome. As such, while we do not know the number of users of the VRA (GDPR and anonymity considerations meant that contact was strictly kept between the individuals and the rehabilitation assistant), we can safely assume that there would have been a greater take-up of this expertise had it been promoted more widely and more thoroughly for longer.

## 4. Discussion

### 4.1. Findings

The aim of this project was to work with social workers from LAs across the UK to co-create, implement, and evaluate a series of mental health and well-being initiatives via a smartphone application. Through the co-creative process (semi-structured interviews and focus group discussions with frontline staff and steering group discussions with senior organizational representatives), a number of psychoeducational, communication, information-sharing, and individual well-being interventions were developed and implemented across the participating social work-employing LAs.

Psychoeducational approaches (i.e., approaches that seek to improve acquisition of knowledge, competencies, and skills) have been demonstrated to be impactful as stress-reduction techniques [45]. Two distinct psychoeducational approaches were developed within this project. The first, accessible through both the app and toolkit, outlined the signs and symptoms of stress and poor psychological well-being, as well as organizational support mechanisms (such as occupational health and employee assistance program) for those who identify signs and symptoms either in themselves or others.

A second psychoeducational approach was the sharing of well-being-related events that are offered through the LA. Research has demonstrated that organizations that have a transient or geographically diverse workforce (such as social workers who are based outside of the main organization and social workers who are often out on visits to vulnerable service users and families) often struggle to share the well-being events that are available for their employees (e.g., [36]). Through the app, therefore, users were notified using a dedicated events page and push notifications as to all new well-being events that occurred within their organization.

Communication (both top-down and bottom-up strategies) is often described as a distinct organizational stressor in H&SC [46]. Therefore, improving communication has the potential to improve both working conditions and psychological well-being. The app allowed individuals to (both anonymously and non-anonymously) feed improvements to organizational procedures and job roles to senior management. These suggestions would also see senior management respond to a selection of these suggestions and respond via all-staff communications, thus enabling both top-down and bottom-up communication strategies.

Finally, the app provided access to a trained and dedicated VRA. The VRA could be contacted through the app to support those at risk of stress-related work attrition and mental health-related sickness absence by providing job-related skills’ support, changes to the work environment, and other occupational coaching [47]. A vocational advice service has been demonstrated to be useful in MH and musculoskeletal cases [48]. We could not, however, determine the efficacy of use or how widely the VRA was used because of data-protection and ethical considerations.

Pre- and post-intervention surveys demonstrated that across the life of the project, the way in which change is communicated within the organization was significantly improved for those who used the app post-intervention. While we cannot definitely determine that these changes were due specifically to the current project, the specific interventional focus on communication improvements likely played an important part. However, while no significant improvements were found in six of the remaining working conditions or well-being, comparison of percentile scoring showed improvements between baseline survey respondents and those at follow-up who were aware of and used the app. As such, these improvements were found in the working conditions of demands, control, managerial support, and role. Although well-being did not significantly reduce following use of the app, well-being was worse for those who were not aware of the app versus respondents completing the pre-evaluation forms and those completing the post-evaluation forms post-intervention. More research is needed to determine if well-being and working conditions improve following use of the app with a larger sample and matched participants over time.

Qualitative interviews that were aimed at listening to the improvements and positives associated with the app and toolkit found the psychoeducational and communication elements of the app particularly helpful. Therefore, the app being a ‘one-stop shop’ for all well-being offerings available through the organization and a single source demonstrating when and where all well-being-related events were happening were deemed particularly useful. This is similar to Ravalier et al. [36], who found through an Appreciative Inquiry methodological approach to developing well-being initiatives, that organizations that are geographically spread (such as local council, health, and social care organizations) found it difficult to share information about well-being events that were happening across different geographical areas. However, the dissemination strategy could be improved to widen possible take-up and participation with both the study, app, and toolkit. This could also improve the trust and acceptance of the VRA’s role as an impartial and external part of the project. This could be improved by members of the research team embedding themselves into participating organizations, and thus becoming ‘insiders’ while maintaining the role of the researcher [49].

### 4.2. Limitations and Strengths

There are a number of limitations associated with this project. Firstly, it cannot be ascertained that this project alone led to significant improvements in working conditions and well-being. Organizational interventions are complex, and a pre-intervention post-intervention survey does not take all of these factors into account. Relatedly, a non-randomized, non-matched pairs, and non-Randomised Control Trial methodology undertaken within this project means that it could not be determined that any changes (or, indeed, a lack of changes) in working conditions and well-being were due to the interventions co-developed with social workers. An important limitation to consider within this project is the worldwide COVID-19 pandemic, which began to take hold partway through the project, with the whole of the UK in lockdown and working from home, although some social workers had to continue work at least part-time from their main place of work. This would likely have had some impact on results and post-intervention responses. For example, a self-selecting sample may have meant that those who required the most support were those who completed post-intervention data collection or, vice versa, those who undertook the post-intervention survey had adequate levels of psychosocial health and working conditions. Furthermore, no attempts were made to determine the efficacy of the VRA. Due to data protection and ethical considerations, no data were collected regarding the VRA’s work within the project or the impact that this work had on the individuals that they worked with.

There are some strengths with respect to this project, too, however. First of all, the project was co-designed with social workers in order to support their own psychological health and well-being. Relatedly, the interventions developed were shown to be applicable across organizations. Due to the differences in organizational type (e.g., rural versus city-based) and geographic location, it is likely, therefore, that the interventions are applicable beyond these organizations among the wider social work-employing LA organizations across the UK.

### 4.3. Future Research

Building upon the strengths and limitations of the project, future research should look to and improve the evaluative approach taken and broaden the scope. Firstly, a randomized, controlled, waitlist trial, in which a selection of social work-employing LAs are randomly allocated to either an intervention group or control group as well as individual matching of responses across time, would support the determining of causality within the project. Furthermore, social workers are increasingly working away from the main organizational hub due to COVID-19 and, thus, may struggle to access organizational support structures and peer support. Therefore, a broadened app that incorporates both the psychoeducational components and an approach that improves peer support (for example, by developing an organizationally specific social media platform) could be particularly useful. Furthermore, by broadening the initiatives available to a nationwide set of approaches, we could support the whole social work community across the UK. The JDR has been demonstrated to be a useful model in the development of work-related psychological difficulties. Utility and inclusion of the JDR in future, long-term evaluation, focusing, therefore, on developing and evaluating interventions that act upon the demands and resources experienced at work, could be an important element of evaluation.

## 5. Conclusions

In summary, over the past 18 to 24 months, we worked with colleagues from across a number of LAs from across different regions of England to develop a series of well-being initiatives for social workers in these authorities. A series of individual interviews and focus group discussions were used to develop interventions that focused around psychoeducation (i.e., awareness of stress and mental health at work), awareness of well-being events, communication, and access to a VRA. Evaluation of the efficacy of the study was undertaken by a set of pre- and post-intervention surveys and post-intervention interviews. Findings suggested that organizational change communication was significantly improved across the project and there were improvements (although not statistically significant improvements) in five of the other working conditions measured.

This project has demonstrated that—despite a global pandemic (COVID-19) impacting proceedings—co-created well-being initiatives for the improvement of mental health and well-being of frontline social care workers have the potential to improve the working conditions of these key frontline workers.

## Figures and Tables

**Figure 1 ijerph-17-08730-f001:**
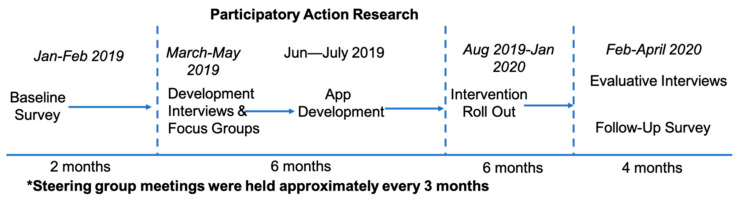
Timeline and methodological steps of the project.

**Table 1 ijerph-17-08730-t001:** Descriptive statistics and percentile scoring for the seven working conditions and the General Health Questionnaire.

Variable	Baseline or Follow-Up	Mean (SD)	Percentile
**MSIT**	Time 1 Baseline	3.66 (0.55)	50th
	Time 2 Follow-up aware of HOW-APP	3.88 (0.45)	95th
	Time 2 Follow-up not aware of HOW-APP	3.70 (0.52)	50th
**Demands**	Baseline	3.17 (0.85)	10th
	Follow-up aware of HOW-APP	3.49 (0.78)	50th
	Follow-up not aware of HOW-APP	3.21 (0.88)	10th
**Control**	Baseline	3.48 (0.70)	50th
	Follow-up aware of HOW-APP	3.66 (0.63)	75th
	Follow-up not aware of HOW-APP	3.46 (0.70)	50th
**Managerial Support**	Baseline	3.72 (0.85)	75th
	Follow-up aware of HOW-APP	4.02 (0.72)	95th
	Follow-up not aware of HOW-APP	3.89 (0.70)	95th
**Peer Support**	Baseline	4.04 (0.63)	95th
	Follow-up aware of HOW-APP	4.17 (0.49)	95th
	Follow-up not aware of HOW-APP	4.08 (0.59)	95th
**Relationships**	Baseline	4.39 (0.72)	50th
	Follow-up aware of HOW-APP	4.33 (0.88)	50th
	Follow-up not aware of HOW-APP	4.31 (0.86)	50th
**Role**	Baseline	4.05 (0.70)	25th
	Follow-up aware of HOW-APP	4.27 (0.62)	75th
	Follow-up not aware of HOW-APP	4.08 (0.64)	25th
**Change**	Baseline	2.92 (0.86)	25th
	Follow-up aware of HOW-APP	3.42 (0.79)	90th
	Follow-up not aware of HOW-APP	3.04 (0.77)	50th
**GHQ**	Baseline	15.35 (6.45)	N/A
	Follow-up aware of HOW-APP	14.43 (6.67)	
	Follow-up not aware of HOW-APP	17.39 (6.81)	

Percentile scoring based on comparison with UK national average, according to Edwards and Webster [27]. Higher scoring is indicative of better outcomes.

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
