# Peer review of "Co-Creating and Evaluating an App-Based Well-Being Intervention: The HOW (Healthier Outcomes at Work) Social Work Project"

_ijerph, 2020, doi:10.3390/ijerph17238730_

Round 1
Reviewer 1 Report
Dear authors,
I congratulate you for an interesting, fairly well-written paper. I have a few recommendations to strengthen it.
First, I suggest you move the aims and research questions to the front end, perhaps at the end of the introduction or create a subsection separate from "child and family social workers" that is aim of the project and contributions. Your paper would be more appealing if I could see your aim, questions and how those contribute to the closing a gap in the literature.
Second, the section "Evaluative Materials" , or what I would call "instruments and scales" is hard to follow. I'd separate the outcome (dependent variable) and independent variables.
Third, in the analytical approach section you mention a few details about the sample size and having an imbalanced pre-post sample, but no numbers are discussed. Given that you mention a correction for small sample sizes and that you have a really high attrition, it would be good to discuss up front attrition between pre and post, the total matched sample size, and the whether your analysis was only performed on the matched sample, on the unmatched (and unbalanced) sample or on both.
Fourth, in the "Results" section when you talk about restricting the analysis to a "smaller sample" you mean only looking at the pre-post matched individuals? If so, you should state it more clearly.
Fifth, given the large attrition you have in pre/post (from 503 to 154) you should have a section that discusses the reasons and implications. In the "Limitations" section you mentioned the COVID-19 pandemic having a large influence on the reduced post sample size, but there is no discussion of the implications. You might have a compositional sample problem or self-selection. For example, those who needed more help remained in the post sample and your results are show larger gains, hence you show upper bound estimates, or those who remained in your post sample are those that were doing OK and your results are smaller and show lower bound estimates.
Sixth, I found a few copy editing errors. Review the manuscript one more time carefully and correct them (e.g. line 479).
Seventh, Add month/year markers to your figure 1, it would make it easier for readers to follow through.
Reviewer 2 Report
Dear authors,
Thank you very much for this article, which gives an important contribution to the current development towards online or app-based health interventions.
However, in my opinion, there is still room for improvement in the first half of the article, which clearly highlights the results of the Participatory Action Research (PAR) approach. Especially what the results of the interviews and focus groups mean for the development of the App(s). Otherwise, the Research Question 1 will not be adequately covered in the article.
Comments in detail:
The citation style does not correspond to the journal's author guidelines. Please adjust this accordingly throughout the text: https://www.mdpi.com/journal/ijerph/instructions#back
Line 2 and following: Are the references published by the authors marked as "anonymous"? It is unusual for references to be anonymised. Unfortunately, this does not make it possible to verify the statements.
Line 37 and 39: You cite a reference 2 times in one sentence. The (2020) in line 37 can be deleted.
Lines 33-42: COVID-19 is only taken up again in the Limitations. No reference is made in the results section. It would be logical, however, if the effects of the COVID-19 pandemic were also reflected in the interviews of the evaluation, or if this was discussed there.
Line 43 and following: The job demands-resources model stands on its own here and makes no reference to social workers. It could be integrated into chapter 1.3 Working Conditions and Stress. Especially because there is a concrete connection with the Management Standards working conditions model in lines 89-91.
Line 106: Please cite the specific source here. The report (HSE,2019) itself only refers to the overview of the LSF Tables, but the text does not deal with sickness absence.
Line 154: „and experience of the experience“ is that what you mean?
Lines 195-197: The content of the sentence on MSIT is very similar to that in Lines 99-95.
Line 215: The specification of the duration of the evaluative interviews is already a result and should be reported in the corresponding part of the results.
Lines 253-258: This is part of the description of the methodological approach. The concrete results of this process should be reported here.
Lines 259-299: These are interview statistics, where are the content-related results? Which of the results were selected for the app? Which themes were identified? Lines 219-220 state the objectives of the analytical approach of the qulitative interviews: identification, analysis, description, and reporting of themes found across a series of interviews. But it should be exactly these topics that play a decisive role in the (co-)design of the apps. Otherwise the PAR approach is not clear here. There is no comprehensible structured overview of the interview results and according to which criteria specific points/ topics were included in the app(s). In addition, all areas of the app should be presented and how they reflect the distinct factors of the Management Standards working conditions model and other aspects. For example, did the social workers twitter (Line 257) wish to exchange information? What was the reason for offering a Vocational Rehabilitation Assistant (Line 293)? On the other hand, this chapter lacks a clear separation between the technical challenges and their consideration in app design, i.e. access restrictions, implementation of push messages etc., and the content structure based on the survey/interview results.
Line 295: Please write out GDPR once.
Line 322 and following: The interventional evaluation is well described and comprehensible.
Line 486: The connection between demands and controls on the development of burnout and the influence on wellbeing, which was introduced at the beginning, should be examined in connection with the future evaluation of the long-term effectiveness of the app(s)/toolkit developed.
Reviewer 3 Report
The problem identified by the authors is an important one. The structure of PAR research to develop interventions to solve problems within communities is very important and even critical to getting the intervention right. It seems that the feedback from those who used the WHO app was generally positive and is timely given the extra stress that workers in this sector are under during the current health crisis. Overall the intervention seems rather achievable across more social work and public health-related settings. Overall, the content of this paper is good, there are just some points of clarification that need to be made.
General Comments
- It is unclear whether this approach and specific technology developed would or could be translated to other fields quickly to help other workforces who suffer as well. A bit more discussion about this would be helpful. This unknown is partly due to the nuances in the specific workforce targeted in this research but also partly because this manuscript is not completely transparent and lacks critical information regarding the App development that would hinder quick reproducibility and research. If possible, more information about the specific questions asked on the surveys (maybe include them in the supplement), the questions asked during the interviews, and the App (how it looks, who made it, how much time it took, how one could access it or the makers if they wanted to develop something similar, etc.) should be included.
- The terms Time 1 and Time 2 survey imply that they are related to some specific something in time when rather they are just baseline and follow-up surveys. I think it’s clearer to say the latter so readers don’t have to keep looking back to determine what was meant by the term.
- Table 1: I would consider changing the “Time 1 and Time 2” to baseline and follow-up or something more descriptive. Conventionally it’s easier to read when words inside of tables are not centered but rather justified left. It would also be helpful to include a reminder in the footer or somewhere else on the table that higher values are worse or better, whichever they are.
- It’s quite unclear what the actual population used in different aspects of this study, both qualitatively and quantitatively are. It would be helpful if some kind of chart or flow diagram could help summarize that information in a holistic and visual sense.
Abstract:
- Please put (PAR) after Participatory Action Research on line 7.
Introduction:
- Line 28 – 30. This sentence seems backward to the thesis. Could use rewording.
- Section 1.2, lines 74-77: The references removed for anonymity don’t match the same words as those reported in the abstract.
- Section 1.4, lines 114-115: This statement is unclear. Is it caseworkers that work on average 11 hours more per week than contracted - more than any other job title in the social sector? Because saying social workers work …. any other social work role… is not clear.
- Section 1.4, lines 117-118: I think that the sentence should end with “with and for UK social workers” to be clear on what is meant by “co-design”. Or perhaps “develop” rather than “co-design”
Methods and Materials
- Section 2.2.1 first paragraph: This part was not clearly illustrated in Figure 1. It is unclear if this first paragraph is referring to the “Time 1 Survey” or something else. Please use consistent language when referring to the surveys, interviews, etc. for clarity. Further, there might be some worth in elaborating on Figure 1 to indicate at what points “Qualitative” data were collected versus “Quantitative or Evaluative” or more specifically the figure could include what exact data points were collected at each time.
- Line 177-179: I am interpreting this line to mean that interviews and focus groups were with a random selection of individuals that may or may not have differed from one another. Is this the correct assumption? This should be made clearer.
- Section 2.2.1 last paragraph: This paragraph is confusing. The first part states that” steering group discussions were with senior social workers and colleagues from across the seven LAs” but the last sentence says they “took place between at least one member of the research team, and two organizational representatives (at least one senior social worker)”. To me, these sentences seem to contradict each other. If that is not the case, please make this clearer.
- Section 2.2.2 lines 193-195: I am unclear by what is meant by this sentence.
- Section 2.2.2 line 199-200: Is 1 or 5 higher (better, etc.) for each of the components in the scale. Please add. Some of the scale parts rank order of what is better or worse as it relates to working conditions may seem intuitive, but to persons not in the field, they may not be intuitive.
Intervention Evaluation
- Was the App available for download on all smartphones (iPhones, Androids, etc.)?
- Line 342: How was “working conditions” combined? Maybe add the combined descriptive statistics to Table 1 for more clarity.
Limitations
- Line 476: “and many social workers within our sample” – it is unclear what is being said by this. Does this imply that the sample included others beyond the social work sector? This connects back to General Comment #3.
- Line 479: the sentence isn’t complete… or seems to have not been completed.
Future research
- Please consider reworking the wording of lines 488-491. I think the point of this sentence is to just say that an RCT study design with the same intervention is needed to truly determine the causality of the intervention on working conditions and wellbeing, but that is not immediately clear.
Conclusions
- Line 501 – why mention “and related professional” now, it hasn’t been mentioned before or indicated that there were “other” job titles in the study sample. Again, this connects to General Comment #3.
Reviewer 4 Report
This is an important study in the area of child and family social work where stress levels are very high and technological advances such as support through an app can be very useful. The following suggestions may improve the quality of the manuscript:
- Some of the refences need updating: page 2, line 66: the reference about cost of absenteeism based on European Commission figures dated 2002 needs updating. Moreover, on the next page, page 3, lines 106-110, a reference should be provided for the claim that "social workers have among the worst working conditions... and job satisfaction." Where is the evidence for this? Please provide a (recent) reference to backup this claim.
- Then on page 3 line 114, it is stated "social workers work on average...more than any other social work role...". This is quite confusing, can you explain what type of "other social worker role" do you mean here? Do you mean a managerial role within family and social work services or do you mean adult social workers or something else? Please clarify.
- Page 5, line 207. Please provide the correct refence for the GHQ (e.g. the original paper is Goldberg D, Williams P. A User’s guide to the general health questionnaire. Windsor: NFER-Nelson; 1988. Not the paper referenced by El-Metwally et al., 2018). Then update the references for studies using the GHQ in an occupational setting, there are more recent studies than Banks et al., 1980 and the El Metwally et al., 2018 could be perhaps cited here as a recent study among others that have used the GHQ-12.
- The evaluation of the app is very limited and not just because of the current circumstances (i.e. the pandemic) where workers are no longer in their offices but also because of the very small number of interviewees taking part in the evaluation procedure and the fact that there is no evaluation or example provided of the kind of response given by management to suggestions for role improvement. Can you provide some examples? I think it would be necessary to add in the limitations section that this was not a full-scale evaluation but rather a small-scale (almost pilot) evaluation and a full evaluation should be designed and implemented in the future.
Round 2
Reviewer 2 Report
Thank you very much for correcting the article.
As it stands, I have no further objections to publication.